# The answer is blowing in the wind: seasonal hydrography and mixing of the inner sea of Tierra del Fuego, Southern Patagonia

Manuel I. Castillo<sup>2,8</sup>, Constanza Zuñiga<sup>1,6</sup>, Carmen Barrios-Guzmán<sup>1</sup>, Natalia Cisternas<sup>1</sup>, José Garcés-Vargas<sup>4,5</sup>, Mauricio F. Landaeta<sup>3,7,8</sup>, Andrea Piñones<sup>5,7</sup>, Marcela Rojas-Celis<sup>2,6</sup>, Alicia I. Guerrero<sup>1</sup>, Maritza Sepúlveda<sup>1</sup>

Correspondence to: Manuel I. Castillo (manuel.castillo@uv.cl)

Abstract. This study characterizes seasonal hydrography and mixing processes in Almirantazgo Fjord, a sensitive ecosystem in southern Chilean Patagonia. Although estuarine and tidal forcing conventionally explain fjord dynamics, wind stress effects remain less understood in this high-latitude region. The study analyses a comprehensive six-month dataset including a moored time-series of temperature, salinity, and dissolved oxygen, cross-fjord CTD transects, and hydrographic profiles derived from seal-deployed sensors. Observations indicate distinct seasonality, shifting from a stratified water column in summer—defined by low-salinity surface water from glacial melt—to a mixed winter state with significantly reduced vertical stability. The analysis identifies persistent, topographically channelled up-fjord winds as a primary physical driver. By applying the Wedderburn number ( $W_b$ ) and mechanical energy balance calculations, we determined that wind stress perturb the pycnocline ( $W_b > 1$ ); furthermore, wind power input rivals that of estuarine circulation during stratified summer periods. Under such conditions, wind forcing amplifies vertical mixing, modulates the pressure gradient, and supports deep-water oxygenation. First-order flushing time estimates suggest slow deep-basin renewal (greater than 5 months), signalling sensitivity to physical forcing. These results indicate that wind constitutes a primary mechanism regulating the hydrographic structure and biogeochemical function of the Tierra del Fuego inner sea.

<sup>&</sup>lt;sup>1</sup>Laboratorio de Ecología y Conservación de Mamíferos Marinos (LECMMAR), Universidad de Valparaíso, Chile

<sup>&</sup>lt;sup>2</sup>Laboratorio de Oceanografía Física y Satelital (LOFISAT), Universidad de Valparaíso, Chile

<sup>&</sup>lt;sup>3</sup>Laboratorio de Ictiología e Interacciones Biofísicas (LABITI), Instituto de Biología, Facultad de Ciencias, Universidad de Valparaíso, Chile.

<sup>&</sup>lt;sup>4</sup>Instituto de Ciencias Marinas y Limnológicas, Facultad de Ciencias, Universidad Austral de Chile

<sup>&</sup>lt;sup>5</sup>Centro FONDAP de Investigación en Dinámica de Ecosistemas Marinos de Altas Latitudes (IDEAL), Universidad Austral de Chile, Chile

<sup>&</sup>lt;sup>6</sup>Programa de Magister en Oceanografía, PUCV/UV

Millennium Institute Biodiversity of Antarctic and Sub-Antarctic Ecosystems (BASE), Santiago, Chile
 Centro de Observación y Análisis del Océano Costero, COSTAR-UV, Universidad de Valparaiso, Chile

Graphical abstract: Schematic representation illustrating the key findings of the study on the Almirantazgo Fjord. The figure depicts typical summer (T<sub>s</sub>,S<sub>s</sub>,N<sup>2</sup><sub>s</sub>) and winter (T<sub>w</sub>,S<sub>w</sub>,N<sup>2</sup><sub>w</sub>) hydrographic profiles obtained via elephant seal-deployed CTD-SRLDs, alongside a conceptualization of the fjord's along-fjord and cross-fjord circulation patterns. Notice that the vertical scale of the upper layer (h<sub>1</sub>) was exaggerated, to focus on the stratification layer.

#### 40 1 Introduction

45

Fjord basins originate from glacial advance and retreat, processes that produce elongated narrow basins characteristically containing one or several glacial moraines known as sills (Dyer, 2019; Farmer and Freeland, 1983; Geyer and MacCready, 2014; Inall and Gillibrand, 2010; Stigebrandt, 2012). These ecosystems play a key role in CO2 sequestration, a function resulting from the undersaturation of their surface waters (Aalto et al., 2021). Furthermore, the growing utilization and exposure of these systems has led to their designation as "aquatic critical zones" (Bianchi et al., 2020).

In these coastal systems, numerous processes operate on different temporal and spatial scales. This results in a complex combination of forces that determines the observed circulation patterns in regional and small-scale basins. The primary mechanism for the physical dynamics is the along-fjord pressure gradient, generated by freshwater input at the fjord head. This

https://doi.org/10.5194/egusphere-2025-5692 Preprint. Discussion started: 26 November 2025

gradient produces the two-layered, along-fjord gravitational circulation (Hansen and Rattray, 1965; Ribeiro et al., 2004). This pattern is manifested as a residual circulation pattern after several tidal cycles (MacCready and Banas, 2012) and can be substantially modified by wind stress (Guo and Valle-Levinson, 2008), bathymetric and frictional effects, the Earth's rotation (Yang et al., 2015), and drag (McCabe et al., 2006).

A complete description of fjord circulation must consider the cross-fjord dynamics. This cross-fjord pattern is generally weak and consists of an overturning motion known as secondary circulation (Chant, 2002; Lacy and Monismith, 2001; MacCready and Geyer, 2010). The along-fjord wind stress can intensify the estuarine circulation during down-fjord winds) or weaken the surface outflow (during up-fjord winds) (Valle-Levinson, 2010).

Global changes in climatic regimes are well-documented (e.g., IPCC, 2021). Associated alterations in freshwater input (which affects stratification) and wind-stress intensity (which affects mixing) could substantially modify the dynamics of regions such as the inner sea of Tierra del Fuego in southern Chilean Patagonia.

Chilean Patagonia is one of the most extensive fjord regions of the world; the shape of this region was formed by the combined effect of glacial erosion since the Quaternary and the tectonic sinking of the central valley (Aracena et al., 2011). The region is formed by more than 100 thousand km of coastline and 40 thousand islands, which generate a complex system of fjords and channels as unique marine ecological hot spots (Hucke-Gaete et al., 2023; Landaeta et al., 2023). This system of channels and fjords in Chile between 41°S and 56°S has been geographically classified into three zones: the northern (41 - 46°S), the intermediate (46-50°S), and the southern (53-56°S) Patagonia (Pickard and Stanton, 1980). Early studies of the Chilean fjords indicated strong parallels with British Columbia fjords (Pickard, 1971; Farmer and Freeland, 1983).

Observations from the last four decades indicate that the predominant zonal (east-west) winds in Southern Patagonia have strengthened at a rate of 0.2-0.3 m s-1 per decade (Garreaud et al., 2013; Giesecke et al., 2021). This strengthening is consistent with an increase in rainfall at a rate of 200 mm per decade in areas south of latitude 50°S (Garreaud et al., 2013). In the region, the Southern Annular Mode (SAM) is related with the generation of the westerlies (Aguayo et al., 2019). The SAM drives the formation of regional westerlies and has shifted toward its positive phase in response to climate change (Garreaud et al., 2013; Aguayo et al., 2019). This positive phase has been connected to an increase to 30% of the westerly winds since 1950 (Downes et al., 2017). El Niño-Southern Oscillation is another significant interannual climate pattern. During El Niño events, a reduction in freshwater inflow is associated with greater vertical mixing and the advection of oceanic waters into northern Patagonia's fjord systems (León-Muñoz et al., 2018).

The global increase in temperature (IPCC, 2021) suggests a significant alteration in the global heat budget and affects the retreat of ice sheets and mountain glaciers (Church et al., 2013). Interaction between glaciers and atmosphere can generate a






local wind circulation known as katabatic winds. These winds are a common characteristic of fjords adjacent to glaciers where they can exceed 20 m s-1 (Farmer and Freeland, 1983; Stigebrandt, 2012; Spall et al., 2017). Along-channels winds may also advect heat and modulate the supply of warmer waters, thereby promoting glacial retreat (Moffat, 2014). The loss of glacial fields in southern Patagonia has been estimated at 10% (Rivera et al., 2017).

While the effects of density and tidal driven circulation on estuarine dynamics are well-documented (e.g. Farmer and Freeland, 1983; Stigebrandt, 2012; Geyer and MacCready, 2014), wind-driven circulation has received comparatively less attention (Soto-Riquelme et al., 2023). Investigations of wind-driven circulation and its effects on the inner-sea of the Chilean Patagonia have been conducted in specific systems within northern and central Patagonia. These include the Reloncaví fjord (e.g. Valle-Levinson et al., 2007; Castillo et al., 2012, 2017), the Reloncaví sound (Letelier et al., 2011), the inner-sea of Chiloe (Soto-Riquelme et al., 2023; Lindford et al., 2024), the Guafo mouth (Ross et al., 2025), the Moraleda channel (Valle-Levinson and Blanco, 2004), the Aysen fjord (Caceres et al., 2002) and Jorge Montt Glacier (Moffat, 2014). Regional Patagonian wind patterns were analysed by Perez-Santos et al. (2019). At this spatial scale, atmospheric rivers have been identified as important factors controlling the regional wind effects (García-Santos et al., 2025). However, although this region is characterized by intense winds and strong tidal modulation, the effects of the wind on the physical dynamics of the inner-sea of Tierra del Fuego remain poorly understood.

The intermediate zone (46-50°S) receives significant freshwater input within the Penas Gulf, originating principally from glacial discharge. The San Quintin glacier is the primary source, and this freshwater contribution has been reinforced in recent years by greater meltwater from Southern Patagonian Icefield (SPI). The resulting freshwater interacts with the South Pacific Current (SPC), causing its advection out of the Gulf. During summer, these waters are therefore distributed both northward and southward. While the magnitude of these salinity anomalies lessens, they remain detectable, extending south of 50°S and north of 46°S (Cisternas et al., in review).

The Southern Patagonian region is strongly affected by the Cape Horn Current (CHC), which is recognized as a component of the Antarctic Circumpolar Current (ACC) system (Lamy et al, 2015; Wu et al, 2019). Nevertheless, knowledge of the CHC's strength and variability remains limited (Zheng et al., 2023). Observational studies report current velocities exceeding 15 cm s<sup>-1</sup> (Chaigneau and Pizarro, 2005), while nearshore measurements show velocities greater than 30 cm s<sup>-1</sup> (Giesecke et al., 2021). Furthermore, an analysis combining model output and altimetry data calculated a CHC transport of 4.36 Sv. This value is comparable to the transport of the Humboldt Current System between 5°S (1.8 Sv) and 15°S (5.2 Sv) (Chaigneau et al., 2013). More recently, research combining model output and in-situ observational data indicates that the CHC is largely geostrophic south of 49°S. This work also shows no clear seasonal variability with velocities of up to 0.3 m s<sup>-1</sup> in the upper 200 m of the water column (Garces-Vargas et al., in review).


This study investigates the hydrographic response of the Almirantazgo Fjord —one of the southernmost fjords of South America, in Tierra del Fuego, Chilean Patagonia— to freshwater input from glacial melting and to along-fjord wind-stress. To describe the variability of the water column and its response to wind forcing, a six-month dataset was analysed. This dataset combines a time series of salinity, temperature, and dissolved oxygen with in-situ hydrographic data acquired from fixed CTD stations and animal-borne satellite CTD.

#### 2 Material and methods

# 2.1 Study region

The Magellan Strait, situated in the southern Patagonia fjord region, connects the Atlantic and the Pacific oceans. Its internal dynamics govern the hydrography of the surrounding inland sea. Despite a sparse population, southern Patagonia possesses considerable economic significance for Chile. The growth of industries such as aquaculture, oil, green hydrogen, and tourism has altered land use, transport, and increased the ecological pressure on the Magellan Strait, Tierra del Fuego, and the Beagle Channel (Ariztía and Undurraga, 2025; Giesecke et al., 2024).

Figure 1: Study region. a) Map of the Patagonian fjord and channels system. b) Topographic and bathymetric map of the southern Patagonian region, where contours depict depth (negative values) and height (positive values). Geographic features

https://doi.org/10.5194/egusphere-2025-5692 Preprint. Discussion started: 26 November 2025

© Author(s) 2025. CC BY 4.0 License.






referenced in the text are identified, including Magellan Strait (MS), Inutil Bay (IB), Whiteside Channel (WC), Almirantazgo Fjord (AF), Brookes Fjord (BF), Ainsworth Bay (AB), Parry Fjord (PF), and Maria Cove (MC). The Darwin Cordillera is denoted by white-shaded high-altitude areas on the island of Tierra del Fuego. c) Placement of oceanographic stations and mooring deployments within the innermost part of the Almirantazgo Fjord, d) Bottom profile along the Almirantazgo Fjord, with distance originating from Maria cove (the head).

Large freshwater inputs from riverine and glacial sources significantly affect the region (Sassi and Palma, 2017). The low salinity of the Magellan Strait results from a north-eastward flow from the Patagonian Current (Brun et al., 2020). The distribution of these waters is governed by the combined effects of wind, tides, and the Antarctic Circumpolar Current (Palma and Matano, 2012). Previous research indicates that waters from the Atlantic Ocean extend their effect westward through the Magellan Strait as far as Punta Arenas. In contrast, the effect of the Pacific Ocean extends eastward to Carlos III Island, where a shallow sill limits its influence on surface waters (Antezana, 1999; Brun et al., 2020).

An important geographical feature is the Darwin Cordillera (DC) mountain range (54° 30', 69° 30'W), which extends 200 km from west to east along the southwestern peninsula of Tierra del Fuego Island. The DC is the core of the southern westerlies and experiences low precipitation. The region is characterized by a strong W-E gradient (Carrasco et al., 2002; Garreaud et al., 2013; Meier et al., 2018; Izagirre et al., 2024). Near the study region (Fig. 1c), the glaciers of Parry fjord have been relatively stable since 1986, but the Darwin glacier retreated 3 km between 1986 and 2014. In this period, the region's glaciers have lost 10% of their area (Rivera et al., 2017).

The Almirantazgo Fjord (AF) is 75 km long, with a width of 16 km in the northern part and 5 km near its head, located at Maria Cove (MC). The AF axis is 116° from the true north, an approximately East-Southeast (ESE) direction. It contains three major glacial connections: Brookes fjord (BF), the Ainsworth bay (AB), and Parry fjord (PF). At the head of the fjord, in Maria Cove, the fjord is shallow, with a mean depth of approximately 30 m for the first 7 km. The bottom then slopes steeply between Isla Tres Mogotes and the mouth of PF, deepening from 30 m to 200 m in less than 3 km. From PF to BF, the fjord depth is approximately 220 m. The deepest part of this basin is in the Whiteside channel (WC), where the bottom depth is approximately 450 m. This is one of the deepest parts of the inner sea of Tierra del Fuego (Fig. 1).

#### 2.2 Atmospheric data

Meteorological data for the region were obtained from the Dirección General de Aeronáutica Civil (https://climatologia.meteochile.gob.cl/), sourced from airports at Punta Arenas (53.00°S, 70.84°W), Porvenir (53.25°S, 70.33°W), and Puerto Williams (54.93°S, 67.62°W). The dataset included wind magnitude and direction (meteorological convention), recorded at 1-hour time intervals between 1960 and 2025.






Additionally, to characterize the wind patterns, this study uses wind reanalysis data from ERA 5 (<a href="https://cds.climate.copernicus.eu/datasets/">https://cds.climate.copernicus.eu/datasets/</a>). The wind components at 10 m high ( $u_{10}$ ,  $v_{10}$ ) were downloaded in spatial grids of 0.25° x 0.25°. The ERA5 is the fifth-generation ECMWF reanalysis of global climate and weather data. Reanalysis integrates model data wordwide observations into a globally complete and consistent dataset on physical laws (Hersbach et al., 2023).

Wind-stress ( $\tau$ ) was calculated using the bulk formula by  $\tau_{(u,v)} = \rho_a \ c_d \ |U_{10}| \ (u_{10}, v_{10})$ , where  $\rho_a$  is the air density (1.2 kg m<sup>-3</sup>),  $u_{10}$  and  $v_{10}$  are the components and  $|U_{10}|$  is wind magnitude at 10 m high,  $c_d$  is dimensionless wind-drag coefficient. The coefficient  $c_d$  was calculated for each wind component value in the time-series following Yelland and Taylor (1996), for  $|U_{10}| < 6 \ m \ s^{-1}$ ,  $c_d = (0.29 + 3.1 \ |U_{10}|^{-1} + 7.7 \ |U_{10}|^{-2}) \ x \ 10^{-3}$ , whereas, for 6 m s<sup>-1</sup>  $\leq |U_{10}| \leq 26 \ m \ s^{-1}$ ,  $c_d = (0.60 + 0.07 \ U_{10}) \ x \ 10^{-3}$ .

## 175 2.3 Sea Level data and tides model

Sea level data were acquired from four tide gauge stations located within the inner sea of the Magellan Strait and the Almirantazgo Fjord (Fig. 1): Gregorio Cove, Punta Arenas, Puerto Williams, and Ushuaia (Table 1). The dataset was obtained from the Sea Level Station Monitoring Facility (<a href="https://www.ioc-sealevelmonitoring.org/">https://www.ioc-sealevelmonitoring.org/</a>) for the period of January to June 2024. At each station, measurements are recorded at a one-minute interval by a pressure sensor positioned near the seabed.

Additionally, a barotropic tidal model (Egbert and Erofeeva, 2002) was applied to determine the spatial distribution of amplitude and tidal current ellipses in the region (Fig. 3). The model provides complex tidal heights (h) and transport coefficients for 15 constituents at 1/30-degree resolution. The principal harmonic amplitudes ( $M_2$ ,  $N_2$ ,  $S_2$ , and  $K_1$ ) were used to characterize the regional patterns of the barotropic tide in the inner sea of Tierra del Fuego.

# 185 **2.4 Hydrography data**

The hydrography data were obtained using two different complementary methods: conventional CTDOF (AML Metrec X) deployments and CTDF (CTD-SRLD) casts from sensors affixed to Southern elephant seals.

The conventional instrumentation consisted of a CTD AML–Metrec X equipped with Temperature, Conductivity, Pressure, Fluorescence/Chlorophyll, and pH sensors, supplemented by an additional Aanderaa Optode 4831 infrared Dissolved Oxygen sensor. The CTDOF was configured for freshwater settings to permit measurements under low-conductivity conditions, with profiles acquired at a rate of four scans per second. At each oceanographic station, the instrument was activated on deck and subsequently lowered to 1 m below the surface, where it was held for one minute to ensure sensor stabilization. Data acquisition was then performed using an electrical winch to a depth of 200 m. To describe the oceanographic conditions, sampling was conducted at five cross-fjord oceanographic stations (Fig. 1). Data were collected during the austral summer (January 15<sup>th</sup> and 31<sup>st</sup>, 2024) and the austral winter (June 26<sup>th</sup> and 30<sup>th</sup>, 2024). A total of 60 cross-fjord profiles were acquired to characterize the oceanographic conditions near the head of the Almirantazgo Fjord.

Oceanographic conditions in the region were also assessed using CTD-SRLD (Bohemme et al., 2009) and CTD-Fluoro tags (Guinet et al., 2013). The CTD tags possess high accuracy (± 0.005°C for temperature, ± 0.01 mS/cm for conductivity, and 2 dBar ± 0.035% for depth) and were calibrated by the manufacturer prior to deployment. These tags were affixed to Southern elephant seals (*Mirounga leonina*) from the colony located at Jackson Bay, located at the head of the Almirantazgo Fjord (Fig. 1c). *In-situ* temperature and salinity measurements were converted to Absolute Salinity (S<sub>A</sub>), Conservative Temperature (CT) following the TEOS-10 (IAPWS, 2010).



Water column stratification was characterized by the buoyancy frequency (N²), calculated as N² = -g/ $\rho_0$   $\partial \rho/\partial z$ , where g is the gravitational acceleration,  $\rho_0$  is the reference density, and  $\partial \rho/\partial z$  is the vertical density gradient. Additionally, CTD data were used to estimate the energy required to homogenize the water column, expressed as the Potential Energy Anomaly (PEA), PEA = 1/h  $\int_{-d}^{0} gz \, (\rho_p - \rho) dz$ , where h is the thickness for the calculation of average density ( $\rho_p$ ). A high PEA value corresponds to high stratification (Simpson et al., 1981). The Freshwater Content (FWC), defined as the equivalent thickness of freshwater in the water column, was also calculated, FWC =  $\int_{-d}^{0} (s_0 - s)/s_0 \, dz$ , where  $s_0$  is the reference salinity at depth h (Blanton and Atkinson, 1983).

To assess the influence of different water masses, an Absolute Salinity of 31 g kg<sup>-1</sup> was used as an index defining the upper boundary of the Modified Subantarctic Water (MSAAW). This water mass is produced by the combination of surface freshwaters with Subantarctic Water (SAAW), the dominant water mass within the region (Silva et al., 2009; Silva and Vargas, 2014).

# 2.5 Time series: observations and data analysis

Time series data of the water column properties were acquired using two distinct but complementary ways. The first method comprised a six-month (January-June 2024) near-bottom (30 m depth) record of sea level, salinity, temperature and dissolved oxygen. The second method involved short-term (5-day) acquisitions of a fine vertical-scale temperature, salinity and dissolved oxygen during January 2025.

The six-month time series was collected using a CTD WiSens NKE Instruments (<a href="https://nke-instrumentation.com/">https://nke-instrumentation.com/</a>). This instrument was secured within a stainless steel, pyramid-shaped frame (1 m x 1 m, 0.7 m height) positioned on the seabed at 30 m depth near Islote tres Mogotes (54.44°S, 69.05°W). The instrument specifications include a pressure range up to 300 m (0.1% accuracy), a temperature range of the -2°C to 35°C (± 0.02°C), and salinity range 2 to 42 PSU (±0.1 PSU). Dissolved oxygen (DO) concentration was recorded by a PMEL MiniDOT (<a href="https://www.pme.com/products/minidot">https://www.pme.com/products/minidot</a>), which functions via an optical sensor. This sensor has an accuracy of ± 0.3 mg L<sup>-1</sup> and a resolution of 0.001 mg L<sup>-1</sup>. The MiniDOT also contains temperature sensors with a range of 0°C and 35°C ± 0.1°C and its housing is rated for 300 m depth. Both the Wisens NKE and

MiniDOT sensors were affixed in individual stainless-steel frames attached to the main pyramid structure. This entire assembly was lowered to the bottom and remained moored until retrieval at the end of June 2024.

The 5-day time series was acquired from a mooring at 54° 27' 54.11''S, 69° 0' 18.36''W (Table 1) between January 22nd and 26th. This instrument array was designed to measure the vertical gradient of temperature, salinity, DO, and pressure. The array included CTD NKE WiSens and a miniDOT positioned near the surface and near the bottom (30 m depth) to determine the vertical gradients of salinity ( $\partial S/\partial z$ ), temperature ( $\partial T/\partial z$ ), and dissolved Oxygen ( $\partial DO/\partial z$ ). Furthermore, HOBO Water Temperature Pro v2 (<a href="https://shorturl.at/uwbYS">https://shorturl.at/uwbYS</a>) thermistors were affixed at discrete depths (2, 5, 7, 10, 15, 20, and 25 m) to resolve the thermal structure water column during this period.

To characterize the temporal variability and dominant oscillations observed in the data, a Morlet wavelet analysis (Morlet et al., 1982; Torrence and Compo, 1998; Grinsted et al., 2004) was applied to the time series of wind stress, sea level, Conservative Temperature, Absolute Salinity, and Dissolved Oxygen. This method permits the computations of both the dominant modes of variability and their temporal variations (Torrence and Compo, 1998). To quantify the relationship between the wind stress and the other variables, wavelet coherence and phase were estimated following the methodology of Grinsted et al. (2004).

**Table 1.** Location and descriptions of the sea level stations, hydrographic (oceanographic) stations, and time series data used in the study.

|             | Station         | Latitude     | Longitude    | Dates                 | Time     |
|-------------|-----------------|--------------|--------------|-----------------------|----------|
|             |                 | (S)          | (W)          |                       | interval |
| Sealevel    | Gregorio Bay    | 52°35'35.54" | 70° 9'44.23" | Jan 01 – Jun 30 /2024 |          |
|             | Punta Arenas    | 53°10'16.25" | 70°54'17.54" |                       | 1 min.   |
|             | Puerto Williams | 54°55'57.12" | 67°36'29.05" |                       |          |
|             | Ushuaia         | 54°49′1.2″   | 68°13′1.2″   |                       |          |
|             | María Cove      | 54°26'31.87" | 69° 3'6.01"  | Jan 15 – Jun 30/2024  | 10 min.  |
| hydrography | CTD 1           | 54°23'51.9"  | 69°08'10.4"  | summer:               |          |
|             | CTD 2           | 54°24'48.6"  | 69°09'38.9"  | Jan 15 – 31/2024      |          |
|             | CTD 3           | 54°25'45.6"  | 69°10'48.1"  |                       | 0.25 s   |
|             | CTD 4           | 54°26'32.6"  | 69°11'37.1"  | winter:               |          |
|             | CTD 5           | 54°26'50.7"  | 69°11'57.8"  | Jun 26 – 30/2024      |          |
| Ц           | A1              | 54°26'31.87" | 69° 3'6.01"  | Jan 15 – Jun 26/2024  | 10 min.  |


|  | A2              | 54°27'54.11"  | 69° 0'18.36" | Jan 20 – 26 /2025          | 5 min.  |
|--|-----------------|---------------|--------------|----------------------------|---------|
|  | ERA 5 pixel     | 54° 11'' 00'' | 70° 00' 40'' | Jan 01 /2024 – Jan 31/2025 | 1h      |
|  | River discharge | 54° 30' 10''  | 68° 49' 33'' | Jan 01 /2024 – Jan 31/2025 | 30 min. |

The potential for the wind stress to perturb the pycnocline was assessed using the dimensionless Wedderburn number  $(W_b)$  following Geyer (1997), Thorpe (2005) and Inall et al. (2015). The number is defined as:  $W_b = (\tau L) (\Delta \rho g)^{-1} h_1^{-2}$ , where L is the length of the fjord (or the horizontal scale of the wind), g is gravitational acceleration, and  $h_1$  is the depth of the upper layer directly influenced by the wind stress  $\tau$ . In a simplified two-layer model, the density difference  $\Delta \rho = \rho_2 - \rho_1$  where  $\rho_1$  and  $\rho_2$  are the densities of the upper and deeper layer, respectively. According to this formulation, a value of  $W_b < 1$  indicates that the density gradient dominates. When  $W_b = 1$ , wind forcing and buoyancy effects are comparable, and horizontal mixing becomes significant When  $W_b > 1$ , wind-driven effects are dominant (Inall et al., 2015).

## 3 Results




# 3.1 Wind pattern: seasonality of the zonal and meridional components

The 54-years hourly reanalysis dataset was used to establish the wind climatology for the study region (Fig. 2). The results indicate that seasonal winds exhibit high directional consistency. Westerlies are the dominant pattern; south of 54°S in the Pacific Ocean, these winds acquire a slight southerly component, with the airflow generally following the coast. Wind magnitudes intensify near the coast, particularly during the warmer seasons (austral spring and summer), when speed exceeds 8 m s<sup>-1</sup> (Fig. 2a, 2b). During the colder seasons (austral autumn and winter), regional winds are weaker (< 6 m s<sup>-1</sup>), particularly offshore in the Pacific Ocean. Within the inner sea of the Magellan Strait and Almirantazgo Fjord, wind magnitude was low (< 4 m s<sup>-1</sup>), comparable to those observed in the open ocean. Regarding direction, winds within the Almirantazgo Fjord appear aligned with the fjord's axis and generally flow toward the fjord head throughout all the seasons (Fig. 2).

**Figure 2:** Climatology of winds in southern Patagonia. Mean wind vector are shown for the periods: a) JFM (January-March, austral summer), b) AMJ (April-May, austral autumn), c) JAS (June-September, austral winter), and d) OND (October-December, austral spring). Data were derived from hourly 0.25 x 0.25° gridded ERA5 reanalysis data covering the period 1970–2024.

The climatology from the meteorological stations indicates a high consistency with the regional winds described previously (see S1). At Punta Arenas (PA), Porvenir (PV) and Puerto Williams (PW), eastward winds (meteorological westerlies) were predominant (> 25%) during all the seasons. The highest wind speeds (c.a. 15 m s<sup>-1</sup>) were observed at PA and PV during summer and spring. PV was an exception, as northward/southward winds were also recorded there during autumn and winter.

For the Almirantazgo Fjord (AF) wind data were derived from a time series of an ERA5 reanalysis pixel (Fig. 2). At this location, the distribution of magnitude and direction was highly consistent with the regional patterns, showing a dominance of eastward winds in all the seasons. The directional distribution indicates that up-fjord winds (directed towards the fjord's head) are prevailing conditions in AF.


# 285 3.2 Sea level and Tidal influence on the region

Tidal amplitudes recorded by the region's tide gauges (Fig. 1 and Table 2) were primarily controlled by the  $M_2$  semi-diurnal harmonics. The Gregorio station measured an  $M_2$  amplitude of 132.7 cm, a magnitude at least double that observed at the other stations (which ranged from 48 cm to 56 cm) located south of the Magellan Strait. At the Punta Arenas station, the  $M_2$  amplitude was 48.1 cm. In the Beagle Channel, south of the Almirantazgo Fjord, the Puerto Williams and Ushuaia stations recorded comparable  $M_2$  amplitudes of approximately 55 cm. The form factor (F) indicated that all the stations are characterized by a mixed, mainly semi-diurnal tidal regime. The lowest F value (0.33) was observed at Gregorio, whereas at the other stations,  $F \approx 0.6$ .

Figure 3. Regional distribution of the principal harmonic constituents: a) M<sub>2</sub>, b) N<sub>2</sub>, c) S<sub>2</sub>, and d) K<sub>1</sub>. Amplitude (m) is represented by color shading, while phase (°) is shown as contour lines. In the caption, the locations of the First (1<sup>st</sup>N) and the Second Narrows (2<sup>nd</sup>N) within Magellan Strait are indicated.

To quantify the tidal influence in the study region, the barotropic tidal Model by Egbert and Erofeeva (2002) indicated that the northern part of the Magellan Strait and its connection with the Atlantic Ocean showed the highest M<sub>2</sub> amplitude (> 3 m). Amplitudes decreased southward, falling below 1 m off Punta Arenas and reaching approximately 0.5 m inside the Almirantazgo Fjord (Fig. 3). A marked change in amplitude (from high to lower) was observed at the Second Narrow, phase


tends to increase from the Atlantic (10°) to the inner-sea reaching 90° into the Inutil Bay. The entire basin from the Inutil Bay to Maria Cove seems to be in phase at 90°. Other harmonic constituents were substantially smaller than  $M_2$  accounting for < 10 % of the variability explained by the  $M_2$  component (Fig. 3).

# 3.3 Temperature and Salinity patterns inside the Almirantazgo Fjord

The complete CTD-SRLD dataset acquired from Southern elephant seals encompassed extensive areas, including the Pacific and Atlantic oceans and the inner sea of the Magellan Strait and Tierra del Fuego. To specifically address the dynamics of the Almirantazgo Fjord, the present analysis utilized only the CTD-SRLD data obtained from within the fjord. The hydrographic data reveal two distinct regimes for the waters of southern Patagonia. Within the inner channels of Tierra del Fuego, low absolute salinity (c.a. 31 g kg<sup>-1</sup>) was observed in deeper waters (depths > 100 m). Conversely, waters in the outer channels were characterized by salinities exceeding 34 g kg<sup>-1</sup>. These findings indicate that the inner channel waters are more stable (stratified) compared to the mixed conditions prevailing in the outer channels.




Figure 4. a) Locations of the hydrographic profiles of Absolute Salinity (SA) and Conservative Temperature (CT) acquired via CTD-SRLD (Southern elephant seals) and CTD-AML.CTD-SRLD data points within Bahia Inutil, Whiteside Channel, and Almirantazgo Fjord are marked in red. b) CT/SA diagram the complete dataset. Segmented black lines are isopycnals of conservative density anomaly (σ<sub>CT</sub>) in kg m<sup>-3</sup> in. c) Summer and d) Winter vertical sections of CT (upper panel) and SA (lower panel). The thin black line in panel (a) indicates the along-fjord transect. Average profiles of Brunt-Väisälä frequency (N²),
 S<sub>A</sub>, and CT for the entire transect are displayed to the right of the lower panels (c) and (d).

To compare and describe the seasonal patterns of the Almirantazgo Fjord, this study utilized CTD-SRLD data acquired from Southern elephant between January 2024 and March 2025. The analysis was restricted to Conservative and Absolute Salinity measurements collected within the region extending from the Whiteside channel and Almirantazgo Fjord (Fig. 4).

The Conservative Temperature/Absolute Salinity (CT/S<sub>A</sub>) diagram (Fig. 4), derived from 3,308 CTD-SRLD profiles acquired by elephant seals, illustrates the wide hydrographic variability within Almirantazgo Fjord. S<sub>A</sub> ranged from a minimum of 26.195 g kg<sup>-1</sup> near the surface (4 m depth) to a maximum of 31.579 g kg<sup>-1</sup> near the fjord bottom. The 10th and 90th percentiles were 29.566 g kg<sup>-1</sup> and 31.013 g kg<sup>-1</sup>, respectively. The mean S<sub>A</sub> was  $30.495 \pm 0.626$  g kg<sup>-1</sup>, and values remained below 31.6 g kg<sup>-1</sup> throughout the entire season.

CT varied from 4.169°C to 11.171°C. The mean CT was  $7.591 \pm 1.185$ °C, with 10th and 90th percentiles of 6.112°C and 9.174°C, respectively (Fig. 4). The conservative density anomaly (segmented lines, Fig. 4b) extended from 1019.8 kg m<sup>-3</sup> in the surface waters during summer to a maximum of 1024.5 kg m<sup>-3</sup> in near-bottom waters during winter. The mean CT and  $S_A$  profiles (Fig. 4c, d) show pronounced upper-water-column stratification in summer, contrasting with the well-mixed conditions observed in winter.

The coss-fjord transect data were acquired using a CTD-AML Metrec during field measurements in January and July 2024 (Table 1). During the austral summer, S<sub>A</sub> ranged from a minimum of 26.295 g kg<sup>-1</sup> near the surface to a maximum of 30.985 g kg<sup>-1</sup> near the bottom, with a seasonal average of 30.212 ± 0.104 g kg<sup>-1</sup>. Water temperature varies between 6.132 °C and 10.810 °C, with a mean of 7.680 ± 0.029 °C. Additionally, Dissolved Oxygen concentrations ranged from 4.005 to 5.045 mL L<sup>-1</sup>, with an average of 4.422 ± 0.033 mL L<sup>-1</sup>.

Figure 5. a) Location of the hydrographic profiles (Absolute Salinity,S<sub>A</sub>; Conservative Temperature, CT) acquired via CTD-345 AML. b) CT/S<sub>A</sub> diagram for the CTD-AML dataset. Segmented black lines represent conservative density anomaly (σ<sub>CT</sub>) isopycnals in kg m<sup>-3</sup>. Summer and d) Winter cross-fjord sections of S<sub>A</sub> and CT, extending from station CTD1 (northern side) to CTD 5 (southern side). The transect location is denoted by the dashed black line in panel (a). Average vertical profiles of CT and S<sub>A</sub> for the entire transect are displayed to the right of panels (c) and (d).

During winter (July 2024), conditions in the Almirantazgo Fjord (AF) shifted. S<sub>A</sub> ranged from 29.008 to 31.009 g kg<sup>-1</sup> with a mean of 30.783 ± 0.040 g kg<sup>-1</sup>. The water column was colder, with CT between 2.967 °C and 7.743 °C (mean: 7.012 ± 0.060 °C). A thermal inversion, a typical fjord structure, was observed, with colder upper waters overlying warmer deeper waters. DO concentrations vary from 3.856 to 5.247 mL L<sup>-1</sup>, with a mean of 4.282 ± 0.041 mL L<sup>-1</sup> (Fig. 5). The resulting Conservative Density range was 1018.8 kg m<sup>-3</sup> to 1024.5 kg m<sup>-3</sup>. Mean cross-fjord profiles indicated that stratification was salinity-dominated. In summer, continuously stratified conditions were present in the upper 30 m depth layer. In winter, however, the S<sub>A</sub> profile was nearly homogeneous throughout the upper 100 m of the water column.




**Figure 6.** Distributions of Freshwater Content (FWC) and Potential Energy Anomaly (PEA) for summer (red) and winter (blue) shown for the (a, b) along-fjord and (c, d) cross-fjord transects. Relative distance from the fjord head (along-fjord) and from the northern side (cross-fjord) is indicated with segmented lines.

Both sets of hydrographic data (along-fjord and cross-fjord CT and S<sub>A</sub> profiles) were used to estimate the Freshwater Content (FWC) and Potential Energy Anomaly (PEA) for summer and winter (Fig. 6). During summer, the FWC exhibited a gradient, decreasing from the fjord head toward the Magellan Strait. At 2.7 km from the head, which is near Parry Fjord (PF), the FWC was approximately 2.7 m. The along-fjord FWC distribution also presented maxima of 2.4 m at AB (42 km from head) and 2 m at BF, before diminishing to 0.9 m near the connection with the Magellan Strait (Fig. 6a). The along-fjord structure of PEA closely resembled that of the FWC, displaying a maximum of 198 J m<sup>-3</sup> near PF and secondary maxima at AB (c.a. 145 J m<sup>-3</sup>) and BF (c.a. 120 J m<sup>-3</sup>). During winter, fewer data were available for a detailed characterization of the upper layer S<sub>A</sub>. However, sufficient data were acquired to describe conditions between AF and BF (near the Whiteside Channel), In this period, both FWC and PEA were substantially lower than in summer; FWC remained below 1.5 m and PEA did not exceed 50 J m<sup>-3</sup> (Fig. 6b).

The cross-fjord FWC and PEA values were consistent with the along-fjord estimations, as summer values exceeded those recorded in winter. In summer, FWC ranged from 1.8 m (north side) to 2.1 m (south side), displaying a slight tilt toward the southern portion of the transect near PF (Fig. 6c). PEA values reached 200 J m<sup>-3</sup> and exhibited a cross-fjord tilt analogous to





that of the FWC. During the winter, FWC was less than 1.0 m and showed only a minimal cross-fjord tilt to the south. The winter cross-fjord PEA was < 40 J m<sup>-3</sup> a value approximately five times lower than observed in summer (Fig. 6d).

## 380 3.4.1 Temporal variability: Summer to Winter Transition

During the austral summer 2024, instrumentation measuring sea level, temperature, salinity, and oxygen was deployed at a fixed station within the Almirantazgo Fjord, Tierra del Fuego. These sensors were subsequently retrieved during fieldwork conducted at the end of June 2024. The winds-stress time series presented in Fig. 7 covers from January to June, permitting a comparison of its variability with the time series of sea level, temperature, dissolved oxygen, and salinity from the A1 mooring (Fig. 1). The following description is based on the data shown in Fig. 7.

The regional wind pattern indicated that winds persistently blew toward the fjord head, and the zonal (east-west) component was dominant across the seasons (Fig. 2). Therefore, the wind-stress magnitude was analysed to characterize the variability and intensity of the winds within the fjord. The wind-stress magnitude recorded a minimum of 2 x  $10^{-4}$  N m<sup>-2</sup> (August  $6^{th}$ ) and a maximum of  $1.59 \times 10^{-1}$  N m<sup>-2</sup> (July  $23^{rd}$ ); both extremes occurred during the austral winter. The mean wind-stress for the time-series was  $3.09 \pm 2.26 \times 10^{-2}$  N m<sup>-2</sup> and the 90th percentile was  $6.24 \times 10^{-2}$  N m<sup>-2</sup>. Seasonal differences were apparent: the mean summer (December- February) wind-stress was  $3.80 \pm 2.26 \times 10^{-2}$  N m<sup>-2</sup> which was higher than the mean winter (June to August) wind-stress  $2.28 \pm 2.16 \times 10^{-2}$  N m<sup>-2</sup>.

The wind stress time series displayed a combination of high-frequency (c.a. 1 day) and synoptic (c.a. 5 days) oscillations from late January (summer) to mid-April (early autumn) 2024. During this summer and early autumn phase, wind stress events exceeding 0.1 N m<sup>-2</sup> occurred in early February and late March. The wind pattern showed a seasonal transition from mid-April to the end of June. During this late period, oscillations were predominantly synoptic (c.a. 5 to 7 days). No event surpassed 0.1 N m<sup>-2</sup>, although notable events > 0.07 were recorded in the mid-May and mid-June 2024 (Fig. 7a).

The Azopardo river discharge time series for January-June contains a data gap between April 15<sup>th</sup> and May 7<sup>th</sup> (Fig. 7). The discharge was higher in summer (48.231 m<sup>3</sup> s<sup>-1</sup>) and lower during winter (32.760 m<sup>3</sup> s<sup>-1</sup>), with a mean of 40.495 ± 4.470 m<sup>3</sup> s<sup>-1</sup>. The data exhibited a pronounced negative seasonal trend from summer to winter, declining at a rate of -2.42 m<sup>3</sup> s<sup>-1</sup> d<sup>-1</sup>. High frequency variability was evident throughout the record; similar to the wind stress, daily and synoptic oscillations seemed dominant (Fig. 7b).

Figure 7. Time series of, a) wind-stress magnitude; b) Azopardo River discharge; c) sea level; d) near-bottom (30 m depth)
Conservative Temperature (red line) and Dissolved Oxygen (green line); and e) near-bottom (30 m depth) Absolute Salinity
(blue line). Data for panels (c), (d), and (e) were recorded at station A1 in Maria Cove. Segmented lines on discharge,
temperature, dissolved oxygen, and salinity indicate the seasonal trend for each time series. Notice that warmer events were
marked with shading red, transition with shading grey and colder with shading blue.



Sea level (SL) was derived from the pressure sensor of the NKE WiSens, SL height was referenced to the minimum recorded value to isolate amplitude and oscillations. Harmonic analysis indicated the dominance of the  $M_2$  constituent (0.59 m), followed by  $K_1$  (0.31 m),  $S_2$  (0.29 m),  $O_1$  (0.22 m), and  $O_2$  (0.13 m). The form factor (F = 0.593) confirms a mixed, predominantly semi-diurnal tidal regime. This tidal forcing accounted for 76% of the total SL variability. An air pressure time series was not available to calculate adjusted sea level; the residual (non-tidal) variability was clearly of synoptic origin. The maximum tidal range was less than 3 m during spring tides and approximately 1 m during neap tides. Between February and June, 11 spring and 10 neap periods were observed; however, the spring tides from February to late March were less pronounced than those in autumn and early winter (Fig. 7c).

- Conservative Temperature (CT) displayed a pronounced negative seasonal trend of -1.212°C d<sup>-1</sup> from summer to winter (Fig. 7d). The maximum CT (10.23°C) was observed during summer, while the minimum (4.98°C) occurred in winter. The time series exhibited a mean of 7.939 ± 0.771°C. This seasonal signal was evident in the higher mean CT during summer (8.084 ± 0.734°C) compared to winter (7.188 ± 0.855°C). The record was characterized by short-duration thermal events: warmer during summer (February to end of March) and colder during autumn-winter (April to June). Similarly, Dissolved Oxygen DO, showed a negative seasonal trend (-0.069 mL L<sup>-1</sup> d<sup>-1</sup>) over the same period (Fig. 7d). The mean DO was 6.946 ± 0.222 mL L<sup>-1</sup> in summer and 6.831 ± 0.510 mL L<sup>-1</sup> in winter. The time series featured high DO events that were brief (< 2 days) between February and mid-March but became more protracted (> 2 days) during autumn and winter, with extended events (> 1 week) observed from mid-May to June.
- The S<sub>A</sub> time series recorded a minimum of 27.561 g kg<sup>-1</sup> (summer) and a maximum of 30.962 g kg<sup>-1</sup> (winter) during the sampling period (Fig. 7e.). The mean S<sub>A</sub> was 30.028 ± 0.730 g kg<sup>-1</sup> and a distinct seasonal trend (0.557 g kg<sup>-1</sup> d<sup>-1</sup>) from fresher to saltier water conditions was evident between February to June. The mean summer value was 30.092 ± 0.646 g kg<sup>-1</sup> compared to winter mean of 30.459 ± 0.315 g kg<sup>-1</sup>. Although the mean seasonal values were comparable, the variance in summer was four times greater than in winter. The low-salinity conditions in summer were associated with brief events (c.a. 1-2 days) of < 30 g kg<sup>-1</sup> waters, which occurred from late January to late March. From April to late June, these low-salinity events were more protracted (> 2 days), such as those in mid-May and mid-June, but remained above 29 g kg<sup>-1</sup> (Fig. 7e).

# 3.4.2 Temporal Variability: Water Column Response to Variable Wind Forcing

The summer-to-winter time series revealed a pronounced seasonal signal and provided evidence of a wind-driven response in the near-bottom (30 m) waters at the head of the Almirantazgo Fjord (Fig. 7). However, this single-depth dataset was insufficient to resolve the vertical structure of the water column's response. Consequently, the short-term mooring was

deployed in Maria Cove, near the Azopardo river outflow (Fig. 1). The mooring was instrumented to record the surface and bottom (30 m) water column response to variations during contrasting wind intensity.

Figure 8. Short time-series acquired at mooring site A2 in Maria Cove (January 20–26, 2025). Panels display, a) along-fjord  $(\tau_u)$  and cross-fjord  $(\tau_v)$  wind-stress components, b) Azopardo discharge, c) demeaned sealevel, d) density gradient  $(\Delta \rho)$ , e) Wedderburn number  $(W_b)$  and f) Dissolved Oxygen gradient  $(\partial DO/\partial z)$ .

On January 20 and through most of January 22, the ang-fjord ( $\tau_u$ ) and cross-fjord ( $\tau_v$ ) wind-stress components exhibited low magnitudes (< |0.05| N m<sup>-2</sup>). A  $\tau_v$  event (c.a. -0.1 N m<sup>-2</sup>) occurred at the end of January 22nd, marking a transition in the time series. This event separated an initial period of weak winds (before 20:00 local time on Jan 22) from a subsequent period of increased wind forcing (> 0.1 N m<sup>-2</sup>) from January 23 to 25 (Fig. 8a). The Azopardo river discharge was relatively constant at approximately 44 m<sup>3</sup> s<sup>-1</sup> during the weak wind period, decreasing slightly to 42 m<sup>3</sup> s<sup>-1</sup> during the period of stronger winds (Fig. 8b). Regarding sea level, four distinct oscillations were observed during the weak wind period, presenting a clear semi-diurnal signal free of high-frequency noise. This pattern changed with the onset of strong winds, at which point the sea level record became characterized by high-frequency variability. During this latter period, the tidal oscillation was completely disrupted by the wind stress, especially between 00:00 to 12:00 of January 25 (Fig. 8c).

To characterize the changes in water column density,  $\Delta \rho = \rho_2 - \rho_1$  was used, where  $\rho_2$  is density at 30 m and  $\rho_1$  is the surface density. On January 21 and early January 22,  $\Delta \rho$  exceeded 10 kg m<sup>-3</sup>. The gradient subsequently diminished to values near 1 kg m<sup>-3</sup> and remained at this low level until the end of January 24. At that point, where  $\Delta \rho$  increased and was maintained above 10 kg m<sup>-3</sup>. It decreased again to approximately 1 kg m<sup>-3</sup> during the end of January 25 and the beginning of January 26 (Fig. 8d).

- The Wedderburn number (W<sub>b</sub>) was calculated using L= 160 km (the basin length from Inutil Bay to the fjord head) and h<sub>1</sub>= 30 m (the basin depth in Maria Cove). For 96% of the deployment, events W<sub>b</sub> remained < 1. Events where W<sub>b</sub> > 1 were observed on January 23 at 20:00. The highest values, which rose above 2 and reached a maximum of 3.77, occurred on January 24 between 20:00 and 23:00. A final event was recorded on January 25 at 02:00 (Fig. 8e).
- At the beginning of the time-series (January 21 to end of January 23), the vertical Dissolved Oxygen gradient (∂DO/∂z) was maintained with minor variability around -0.05 mL L<sup>-1</sup> m<sup>-1</sup>. Subsequently, the gradient increased to > -0.1 mL L<sup>-1</sup> m<sup>-1</sup> until mid-January 24. Following this, from mid-January 24 to mid-January 25, ∂DO/∂z was approximately 0 mL L<sup>-1</sup> m<sup>-1</sup>. The most negative gradient (c.a. -0.03 mL L<sup>-1</sup> m<sup>-1</sup>) was observed at the end of January 25 (Fig. 8f).

#### 480 3.5 Spectral Analysis: pattern of variability near the bottom

The wind-stress magnitude exhibited high spectral amplitude for the oscillation with periods greater than 24 hours. This diurnal band showed its greatest amplitude from late January until the beginning of May, during the winter, this band was virtually absent (Fig. 9a). The subsequent period was observed in the 3-days band, which displayed high amplitude at the beginning of the time series (January 25 to February 24) and again around March 20 and mid-May. The 10-day band was the most significant period for the wind-stress magnitude. The first two weeks of February was the only time this band showed lower amplitudes; during that period, the variability was centred at 3 days. Furthermore, distinct events with high amplitudes spanning a broad

range of periods (12h to 256h) were centred on specific dates: February 5, March 30, April 20, May 10, June 3, and June 14 (Fig. 9a).

**Figure 9.** Wavelet analysis for the time series acquired at the A1 mooring: a) wind stress, b) sea level, c) Conservative Temperature, d) Dissolved Oxygen and e) Absolute Salinity. For each variable, the wavelet power spectrum is shown (left panels), and the global wavelet spectrum is shown (right panels). In the power spectra, power is color-coded (red = maxima),

https://doi.org/10.5194/egusphere-2025-5692 Preprint. Discussion started: 26 November 2025

© Author(s) 2025. CC BY 4.0 License.

and the black contour represents the cone of influence where edge effects become important. In the global spectra, the dashed line indicates the 95% confidence level. Horizontal dashed lines denote periods of 4, 12, 24, and 72 hours.

The sea level (SL) data were dominated by tidal variability, with the highest spectral amplitudes occurring in the semi-diurnal (12 h) and diurnal (24 h) bands. Both bands exhibited a pronounced fortnightly modulation; this modulation became less distinct in the 12h band from mid-May until the end of the time series. Other evident variability was contained at 4 h band with low amplitude and bi-weekly modulation and the low-frequency (longer than 1-day periods) which showed marked high amplitudes at 3 days (around 72 h) at the beginning of the time-series until middle of February and beginning of April. The most important (and significant) period band was centred at 10 days (256 h) into the synoptic band. This band persists along the entire time-series but diminishes its amplitude between May and June (Fig. 9b).

- The CT data indicated the significance of the 3-day band. High amplitudes were observed in this band during January and February; its amplitude subsequently decreased until June 3<sup>rd</sup> and 14<sup>th</sup>, when two distinct high-amplitude pulses were recorded (Fig. 9c). The global spectrum confirmed the relevance of the synoptic band, centred at 10 days (approx. 256 h). The wavelet power spectrum shows that this 10-day band was less prominent during March and April (Fig 9c).
- The DO wavelet exhibited a pattern analogous to that of CT at the start of the time series, characterized by high variability within the 3-day band. However, unlike CT, the DO data displayed high amplitudes across the 3-day to 10-day bands between March and April; during this interval, the 5-day band appeared to intensify around March 20th and April 24th. Throughout the record, multiple high-amplitude events occurred across various periods (Fig. 9d).
- For S<sub>A</sub>, the time series from January 23<sup>rd</sup> to March 5<sup>th</sup> was characterized by low spectral amplitudes across all periods resolved by the wavelet analysis. Following this, from March to June, high amplitudes developed at periods greater than 3 days. Within the 5- to-10 day, S<sub>A</sub> variability initially exhibited high amplitudes centred at 5 days (late April). From mid-May until the end of the record, the dominant variability shifted to the 10-day band. High-amplitude events spanning all periods > 24 h were observed centred on June 3<sup>rd</sup> and June 13<sup>th</sup>.

#### 520 4 Discussion

The waters of the inner sea of Tierra del Fuego and the Magellan Strait are characterized by low-salinity (c.a. 31 g kg<sup>-1</sup>) in comparison to the adjacent Pacific Ocean and Atlantic Ocean, where salinities are approximately 34 g kg<sup>-1</sup> (Panella et al., 1999; Palma and Silva, 2004; Brun et al., 2020). These oceanic waters enter the inner-sea and mix with regional freshwaters inputs. Strong tides (Medeiros and Kjerfve, 1988) and strong winds are the primary mechanisms driving mixing within the region.

The study of Brun et al. (2020) identifies the hydrographic condition of the Magellan Strait as the source of the low-salinity waters observed in the Atlantic; this study attributes the mixing primarily to tidal forcing. In the atmosphere, the Southern Annular Mode and other hemispheric-scale modes exert a substantial influence on Patagonia weather (Garreaud et al 2009). The region experiences an intensification of westerlies over Tierra del Fuego, which shows a significant positive trend during the austral summer (Garreaud et al., 2013). Local topography often channels these winds down-fjord (Oltmanns et al., 2014). This effect significantly impacts surface waters and ice, which can be driven down-fjord, subsequently promoting air-sea gas exchange and heat, and facilitating deep-water exchange within the fjords (Klymak et al., 2025).

The role of the wind promoting the deep exchange is an important dynamic component in semi-enclosed coastal systems, such as fjords, which are characterized by limited ventilation. Recent studies have documented warming and deoxygenation in the Puyuhuai Fjord (Lindford et al., 2023) and in British Columbia (Jackson et al., 2021). Fjord circulation is dominated by the estuarine circulation—resulting from forcing such as density gradient, tides, and winds—which produces the characteristic two-layered vertical residual pattern of an upper outflow over an inflow (e.g. Stigebrandt, 2012). According to Soto-Riquelme et al. (2023), the role of wind has been understudied compared to density and tidal drivers in fjords; this is also the case for the inner sea of the southern Patagonia fjords. When winds blow up-fjord (opposing the estuarine outflow), this can yield a three-layered residual pattern, particularly in fjords with weak tidal influence (Geyer, 1997).

#### 4.1 Along-Fjord Exchange of Upper and Deep Layer

Utilizing the hydrographic data acquired from two complementary sources (CTD-SRLD and CTD-AML), within the Tierra del Fuego fjord system, and considering the basin geometry, a first-order estimation of the flushing time for the upper  $(T_{F1})$  and deep  $(T_{F2})$  layers of the Almirantazgo Fjord was performed. This approximation, based on the Knudsen theorem, required determining the total fjord volume  $(V_T)$ .  $V_T$  was calculated by approximating the basin as two segments: (1) an averaged width of 12 km, length of 75 km, and depth of 250 m between BF to PF; and 2) a width of 5 km, length of 9.6 km, and depth of 200 m between PF and the fjord head. Based on these geographic features,  $V_T$  was estimated to be 2.35 x  $10^{11}$  m<sup>3</sup>.

The AF was conceptualized as a two-layer system, and considering a long-term balance between precipitation and evaporation. Based on an upper layer ( $h_1 = 30 \text{ m}$ ), the upper ( $V_1$ ) was 2.84 x  $10^{10} \text{ m}^3$ , and the deep volume ( $V_2$ ) was 2.07 x  $10^{11} \text{ m}^3$  ( $V_T = V_1 + V_2$ ). Acknowledging that the upper layer in fjords typically forms a salt wedge (deeper at the head, shallower at the mouth), the transversal area ( $A_1$ ) was estimated using an average  $h_1$  of 5 m, yielding  $A_1 = 3 \times 10^4 \text{ m}^2$ . A phase velocity ( $c_1$ ) of 1.1 m s<sup>-1</sup> was assumed. The upper layer flushing time ( $T_1$ ) was calculated as  $T_1 = V_1 / Q_1$  where the outflow  $Q_1 = A_1 c_1 = 2.2 \times 10^4 \text{ m}^3 \text{ s}^{-1}$ . This calculation resulted in an upper layer flushing time ( $T_1$ ) of 16 days. For the highly stratified summer conditions, the deep layer inflow ( $Q_2$ ) was estimated using Knudsen theorem salt balance ( $Q_2 = Q_1 S_1 / S_2$ ), resulting  $Q_2 = 1.65 \times 10^4 \text{ m}^3 \text{ s}^{-1}$ . This inflow corresponds to deep layer flushing time ( $T_1$ ) of 145 days. In contrast, during mixed winter conditions

(using  $S_1 = 28.5$  g kg<sup>-1</sup> and  $S_2 = 31$  g kg<sup>-1</sup>), the calculated flushing times increased substantially to  $T_{F1} = 30$  days and  $T_{F2} = 238.3$  days.

The calculated upper layer flushing times were longer than those reported for other highly stratified systems in northern Patagonia. For instance, Castillo et al. (2016) determined an upper-layer flushing time of 3 days for the Relocavi Fjord, and Calvete and Sobarzo (2011) found a similar 5-day flushing time for waters between the Guafo mouth and Elefantes Fjord. The deep layer of the Almirantazgo Fjord exhibited flushing times ranging from 5 (summer) to 8 months (winter). These values are shorter than those in other fjord systems, such as the Puyuhuapi Fjord, where Pinilla et al. (2020) reported deep-water residence times exceeding 1 year. However, direct comparison is constrained by methodological differences; Pinilla et al. (2020) employed a second-order approximation using numerical model to determine water age (e.g. Deleersnijder et al., 2001). These results indicate that the upper layer of the Almirantazgo Fjord has extended residence times (approx. two weeks), which may promote the concentration of materials within the basin. To serve as a tool that supports management and decision-making, this first-order estimation of exchange times will require validation by a second-order estimation in future work.

#### 4.2 Freshwater Dynamics and Stratification

Traditional CTD measurements acquired near Jackson Bay during both January and June recorded salinities below 31 g kg<sup>-1</sup>.

Data from the time series mooring (recovered in June) further indicated that near-bottom (c.a. 31 m depth) salinities reached a minimum of 27 g kg<sup>-1</sup> during the austral summer (January). This was followed by a seasonal increase, with values reaching a maximum of 30.5 g kg<sup>-1</sup> near the recovery dates in June 2024. The low salinities characteristic of the inner sea of Tierra del Fuego is attributed to the input of glacial meltwater, which exhibits strong seasonality.

During summer, the along-fjord Freshwater Content (FWC) was higher at distances corresponding to the locations of major glacial areas (Fig. 6a,b). This finding suggests that direct glacial meltwater input is the primary driver of the FWC increase. Although a lack of comprehensive winter data prevents full confirmation, cross-fjord FWC measurements show that summer values are at least twice as high as those observed during the winter (Fig. 6a,b). Spatially, the FWC increased toward the fjord head in the along-fjord transect and toward Parry Fjord in the cross-fjord transect. This distribution is consistent with a surface slope toward the fjord head, which drives a two-layered system (surface outflow and deep inflow). In the cross-fjord transect, a lateral inclination (tilt) is apparently located to the left of the upper outflow (Fig. 6c,d). This cross-fjord tilt suggests the influence of the Earth's rotation. To evaluate this, the internal Rossby radius of deformation (R<sub>i</sub>) was calculated as R<sub>i</sub> = c<sub>i</sub> f<sup>1</sup>, where c<sub>i</sub><sup>2</sup> = g' h<sub>1</sub> and g' = g Δρ/ρ<sub>0</sub>. For the stratified summer conditions, R<sub>i</sub> = 9 km. This value is on the same order as the width of the head sub-basin and smaller than the width of the down-fjord-sub-basin (Fig. 1). This supports the hypothesis that the AF is sufficiently broad for the Coriolis effect to deflect the outflow, confining it to the left (south) side of the circulation.

https://doi.org/10.5194/egusphere-2025-5692 Preprint. Discussion started: 26 November 2025

© Author(s) 2025. CC BY 4.0 License.

615

620

# 4.3 Patterns of Variability near the bottom

Wavelet analysis of the time series indicated a dominance of low-frequency (periods > 24 h) variability in wind-stress magnitude, Conservative Temperature (CT), Dissolved Oxygen (DO) and Absolute Salinity (S<sub>A</sub>). The sea level (SL) was the exception (Fig. 9b), being dominated by the semi-diurnal (12 h) and diurnal (24 h) signals. This tidal dominance is consistent with the harmonics analysis and the calculated form factor (F = 0.593).

The SL time series uniquely displayed a clear 4h signal, which appeared modulated by the spring-neap cycle. This period is not attributable to the  $M_6$  harmonic, which was not a relevant component in the harmonic analysis. A potential source for this variability is a barotropic seiche (natural oscillation) of the basin, like that described by Stigebrandt (1980). For the barotropic case, the phase velocity (c) is  $c = (g h)^{1/2}$ . Using a basin-averaged depth (h) of 220 m and gravitational acceleration (g), thus  $c = 46.4 \text{ m s}^{-1}$ .

The natural oscillation period (T<sub>N</sub>) is defined as T<sub>N</sub>= 4L c<sup>-1</sup> (Rabinovich, 2018), where L is the basin length. Using L = 165 km (from the Magellan Strait connection to Maria Cove), the calculated T<sub>N</sub> is 3.95 h. This result is highly consistent with the 4 h period observed in the sea level wavelet (Fig. 9b). Excitation for this mode typically occurs following the cessation of sustained winds; and the daily wind variability in the region may be sufficient to induce this oscillation and intensify basin-wide mixing (Stigebrandt, 1980). In stratified systems, internal (baroclinic) seiches can also be excited in the pycnocline, contributing to entrainment and enhancing heat and salt exchange between layers e.g. in the Gulmar fjord (Arneborg and Liljebladh, 2001), the Gullmaren fjord (Djurfeldt, 1987) and the Reloncavi fjord (Castillo et al., 2017)). For the Almirantazgo Fjord (AF) the mode-1 baroclinic phase velocity (c<sub>1</sub>) in summer is 1.1 m s<sup>-1</sup>, yielding an internal seiche period (T<sub>N1</sub>) of 7.6 days. Under the less stratified winter conditions, this period extends to 14.3 days. Future studies incorporating water column current data, will be required to confirm the influence of internal seiche on mixing processes.

Up-fjord winds generate a surface slope  $(\partial \eta/\partial x > 0)$ , highest at the fjord head. In stratified conditions  $(\Delta \rho >> 1)$ , this results in a deepening of the pycnocline at the head, creating an opposing pycnocline tilt  $(\partial h_1/\partial x 




The response is a deepening of the pycnocline at the head and shallowing at the mouth (Fig. 4c). This salinity structure is 625 concurrently modulated by summer glacial meltwater (Fig. 6a). This freshwater forms a buoyant plume. Under weak winds, this plume would likely be deflected by the Coriolis force and exit the fjord. However, moderate to-intense up-fjord winds acting within the baroclinic adjustment time ( $t_s = 1.9$  days)—appear to overcome this deflection, trapping the plume at the fjord head. During this process, the plume gains heat. Consequently, summer up-fjord wind events are associated with warmer 630 near-bottom waters (Fig. 7d). This is consistent to the observations made by Aravena et al., (2025) in Punta Santa Ana at the south of the Magellan Strait. The associated turbulence also increases DO and promotes mixing of the upper freshwater, which reduces the salinity of the deeper waters (Fig. 7e). This process is markedly seasonal. A regime occurred in mid-March 2024, after which wind intensity decreased, and strong events became primarily synoptic. In this early autumn period, up-fjord wind events began to be associated with colder waters. In winter, the up-fjord wind events were related to colder waters. During 635 winter, surface water resides longer (t<sub>s</sub> = 3.6 days) at the surface than in summer and thus loses heat to colder overlying air. These colder events are associated with high DO, resulting from enhanced turbulence (oxygen ingress) and the higher solubility of oxygen in colder water (Fig. 7d). Observations in the short-term are consistent with the synthesis. Following a strong upfjord wind pulse (Wb> 1), the DO gradient approached zero ( $\partial DO/\partial z \sim 0$ ), implying oxygenation of the deeper layer (Fig. 8e,f). The time-domain analysis (Fig. 7) retained the linear trends to reflect the seasonality. The decreasing river discharge 640 trend and increasing salinity trend reflect the freshwater impact. However, the FWC estimations (Fig. 6a) clearly indicate that freshwater inputs from glacial locations are more significant than riverine inputs. Although the specific volume of glacial inputs was not estimated, this study concludes that the combination of this substantial freshwater input with tidal and winddriven mixing maintains the low-salinity conditions of the inner-sea of Tierra del Fuego. Here, near-bottom waters register  $S_A < 32 \text{ g kg}^{-1}$ , distinct from adjacent open ocean waters ( $S_A > 33 \text{ g kg}^{-1}$ ).

# 4.3 Wind influence on the inner sea of Almirantazgo Fjord

While ERA5 reanalysis data are valuable for assessing the spatio-temporal variability of the wind forcing in southern Patagonia, comparisons with *in-situ* meteorological stations (e.g. Punta Arenas and Porvenir) reveal that ERA5 magnitudes are typically 20% lower than observations. This discrepancy suggests that the maximum wind-stress events within the AF may be greater than the estimates presented in Fig. 7 and Fig. 8.

To assess the potential for wind-driven mixing, the dimensionless Wedderburn number  $(W_b)$  was calculated. A characteristic summer wind-stress ( $\tau = 0.17 \text{ N m}^{-2}$ ), a typical density difference ( $\Delta \rho = 3 \times 10^{-3} \text{ kg m}^{-3}$ ), an upper layer  $(h_1 = 30 \text{ m})$ , and the basin length (L= 165 km) yields  $W_b = 1.01$ . This value indicates that wind events can perturb the pycnocline. Observational data from the short-term deployment (Fig. 8e) confirm this: during highly stratified conditions,  $W_b$  repeatedly exceeded 1 and reached values as high as 3.7, demonstrating that wind is sufficient to drive dynamics at the fjord head.

https://doi.org/10.5194/egusphere-2025-5692 Preprint. Discussion started: 26 November 2025

© Author(s) 2025. CC BY 4.0 License.





EGUsphere Preprint repository

The time series (Fig. 8) also reveals a feedback mechanism. Strong winds mix the upper water column, which diminishes the density gradient ( $\Delta \rho$ ). This reduction in stratification, in turn, increases the  $W_b$  (as  $\Delta \rho$  is in the denominator), enhancing the wind's mixing efficiency. Conversely, when winds weaken (due to synoptic and daily oscillations), freshwater discharge strengthens the density gradient. This re-stratification requires greater wind energy to overcome, and in the absence of strong wind,  $W_b$  returns to near-zero values (Fig. 8e).

To further quantify the wind's role, the power per unit area generated by the wind ( $d\phi_W/dt$ ) available for mixing was calculated, following methodologies from Denman and Miyake (1973) and Bowden (1981). Using typical summer values ( $\rho_a = 1.23$  kg m<sup>-3</sup>;  $\rho=1020$  kg m<sup>-3</sup>,  $c_d=1.1x10^{-3}$  and a strong wind intensity  $W_{10}=15$  m s<sup>-1</sup>), the estimated wind power is  $d\phi_W/dt=5.5$  x10<sup>-3</sup> W m<sup>-2</sup>. This value was found to be similar for winter conditions.

To determine if this wind power is sufficient for mixing, it must be compared to the power required to maintain stratification  $d\phi_E/dt$  by the estuarine circulation (Simpson et al., 1990). This stratification power is given by:  $d\phi_E/dt = (1/320) (g^2 h_1^5/A_z \rho) (\partial \rho/\partial x)^2$ . The vertical eddy viscosity (A<sub>z</sub>) was estimated as A<sub>z</sub>= 0.2( $\epsilon$ /N<sup>2</sup>), using an observed turbulent dissipation rate ( $\epsilon$  = 7x  $10^{-5}$  W kg<sup>-1</sup>) from a recent survey (Rojas et al., in review) and the maximum summer buoyancy frequency (N<sup>2</sup>= 0.0158 s<sup>-2</sup>). This calculation yields A<sub>z</sub>=1.3 x10<sup>-3</sup> m<sup>2</sup> s<sup>-1</sup> and a corresponding stratification power of  $d\phi_E/dt$  = 5.0 x10<sup>-3</sup> W m<sup>-2</sup>. In contrast, the weaker winter stratification (N<sup>2</sup>= 0.0011 s<sup>-2</sup>) results in a much lower power requirement ( $d\phi_E/dt$  = 3.5 x10<sup>-4</sup> W m<sup>-2</sup>). These first-order approximations indicate that the wind power ( $d\phi_W/dt$ ) is comparable to the estuarine stratification power ( $d\phi_E/dt$ ) during summer. This suggests that even when the water column is highly stable, the wind is strong enough to induce mixing.

Despite the dominant tidal forcing in the Magellan Strait (Fig. 3) and the strong salinity-driven stratification from glacial melt (Fig. 4c, 5c, 6a, 6c), the wind is a critical mixing agent. Both the Wedderburn number analysis and the mixing power comparison confirm the wind's capacity to mix the upper water column. This study provides the first quantification of the wind-driven effect in the southernmost fjord of Chile (Fig. 10).

In the context of climate change and the projected intensification of the Southern Hemisphere westerlies, these findings are critical. The Almirantazgo Fjord ecosystem—characterized by slow deep-water renewal and mixing processes dependent on both tides- and wind—will be profoundly influenced by future atmospheric changes. Therefore, while tidal and buoyancy forces establish the baseline conditions, the answer to the fjord's physical and ecological future is unequivocally linked to the wind.

# **5 Conclusions**

The hydrodynamics of the inner sea of Tierra del Fuego are primarily controlled by the seasonal cycle of glacial freshwater input, which establishes strong stratification during summer ( $N^2 \approx 0.0158 \text{ s}^{-2}$ ) and modulates basin renewal rates. This buoyancy forcing drives a classic two-layer circulation characterized by rapid upper-layer flushing-time (16 days) but slow deep-water renewal, which extends from 145 days in summer to 238 days in winter. While M<sub>2</sub> tidal energy dominates the adjacent Magellan Strait, our analysis confirms that the specific dynamics of the Almirantazgo Fjord are governed by the interplay between this glacial freshwater buoyancy and along-fjord wind stress.




690

A key contribution of this study—supported by unprecedented spatial coverage from instrumented southern elephant seals is the quantification of wind-driven mixing. Even under highly stratified summer conditions, wind power  $d\phi W/dt \approx 5.5 \text{ x } 10^{-3}$ W m<sup>-2</sup> was found to be comparable to the stratifying power of the estuarine circulation. The frequent occurrence of Wedderburn numbers > 1 (peaking at 3.7) provides direct evidence that synoptic wind events are sufficient to destabilize the pycnocline. Additionally, a persistent 4-hour sea level oscillation suggests the excitation of a barotropic seiche, identifying a secondary mechanism for interior mixing.

Ultimately, these physical processes have critical biogeochemical consequences. The observed oxygenation of near-bottom waters following strong up-fjord wind pulses demonstrates that atmospheric forcing is essential for ventilating the deep basin. As climate change projects an intensification of the Southern Hemisphere westerlies, the physical and ecological future of this fjord system will be increasingly defined by its sensitivity to wind-driven mixing.

# Appendix A

**Figure A1.** Regional wind climatology for the 1970-2024 period. Wind direction follows the oceanographic convention (indicating the direction towards which the wind blows).

**Figure A2.** a) Elephant seal trajectories within the Patagonian fjords and channels during 2024 (red dots) and 2025 (yellow dots). Comparison of CTD-SRDL b) salinity and c) temperature data with is situ observations (CIMAR) and GLORYS model outputs for the defined by the rectangle in (a). d) Absolute Salinity of a transect between the Pacific Ocean, the Tierra del Fuego inner-sea, and the Atlantic Ocean.

# Authors contribution

MS, CBG, AIG, MFL and MIC designed the study and wrote the initial manuscript draft. AP and JG-V contribute to improving the subsequent versions of the manuscript. NC, MR and CZ helped with data analysis. Discussions and iterative feedback from all co-authors significantly contributed to the revision of the manuscript.

# Competing interests

The authors declare that they have no conflict of interest.

# Acknowledgments

The authors thank the researchers, students, and field personnel who assisted in data collection; their efforts made this study possible. Field measurements received funding from Anillo Seals ATE220033, with supplementary support provided by RED 21992 (MINEDUC, Chile). Logistic helps on Punta Arenas was carried out by IDEAL-FONDAP 15150003. CTD-AML data acquisition utilized instrumentation funded by FONDEQUIP EQM170115 (MIC). Furthermore, this research received financial support from FONDECYT 1231058, CIMAR 27F 24-11 (CONA, Chile), FONDEF ID22I10206. JG-V received support from FONDAP N° 15150003. This work forms part of the academic portfolio of MIC for full professorship at the University of Valparaíso (UV).

# Code and Data Availability Statement

The datasets used open source and are available online: hourly ERA 5 wind data is available at 750 https://cds.climate.copernicus.eu/datasets/reanalysis-era5-single-levels?tab=download, sealevel data at https://www.iocsealevelmonitoring.org/, tidal model is available by request at https://github.com/chadagreene/Tide-Model-Driver. The wind data from meteorological stations along Chile is available at https://climatologia.meteochile.gob.cl/application/requerimiento/producto/RE3008. In addition all oceanographic data sets 755 will be uploaded at National Oceanographic Data Center of Chile (https://cendhoc.shoa.cl/inicio). All scripts used to obtain the results presented in this study could be shared upon request at the corresponding author.

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
