# Peer review of "The answer is blowing in the wind: seasonal hydrography and mixing of the inner sea of Tierra del Fuego, Southern Patagonia"

_EGUsphere, 2025_

## Referee Comment (RC2)

**General Comments**

The manuscript by Castillo et al. explores the seasonal hydrography and mixing processes of the Almirantazgo Fjord (AF) in Southern Patagonia. This study stands out for its multi-platform approach, particularly the integration of traditional mooring data with autonomous sensors deployed on Southern elephant seals, which provides a significant spatial expansion of hydrographic data in a logistically difficult region.

Thematically, the paper is well-aligned with current trends in fjord oceanography, which have moved beyond purely buoyancy-driven models to recognize the role of synoptic wind events in regulating exchange flow and vertical structure. While the dataset and the research questions are relevant to the scope of *Ocean Science*, the manuscript requires substantial revision to address critical inconsistencies in physical scaling and the generalization of its conclusions. Several key numerical statements are internally inconsistent (units, magnitudes, and parameter choices), which currently limits reproducibility. These issues must be corrected and the full set of inputs reported for each diagnostic.

**Major Comments**

- A primary concern is the selection of the parameters used to calculate the Wedderburn number ($W_b$). The authors use a length scale $L = 165$ km, representing the distance from the Magellan Strait to the fjord head. Given the complex and curved geometry of the Almirantazgo Fjord and its connection to the Whiteside Channel, using the total path length as the effective scale for wind action is physically debatable.

  Moreover, the manuscript defines $h_1$ as the thickness of the upper layer in the two-layer framework underlying $W_b$, yet the calculation adopts $h_1 = 30$ m described as "the basin depth in Maria Cove", this is conceptually inconsistent. Because $W_b \propto h_1^{-2}$, this choice can strongly bias the magnitude of $W_b$ and thus the inferred importance of wind-driven tilting. The authors should (i) clearly state how $h_1$ is diagnosed from the hydrography (e.g., mixed-layer depth, interface depth, or depth of maximum stratification) and (ii) provide a sensitivity analysis showing how $W_b$ varies across plausible ranges of both $h_1$ and $L$ (e.g., using straighter sub-basin lengths and upper-layer thickness estimates). Without this, the claim of wind dominance remains sensitive to a small set of potentially over-scaled parameters.

- The estimation of flushing times ($T_{F1}$, $T_{F2}$) in Section 4.1 is physically inconsistent and lacks a clear methodological basis. The authors must explicitly include the fundamental equations used and define every term. Currently, the explanation is confusing: a layer depth of $h_1 = 30$ m is used for volume ($V_1$), while a drastically different $h_1 = 5$ m is used for the transversal area ($A_1$), all while assuming a phase velocity ($c_1 = 1.1$ m s$^{-1}$) as the scale for mass transport.

  It is not clear why a wave phase speed ($c_1$) is assumed as the advective velocity for this approximation. In a typical Knudsen balance, the conservation of volume and salt is defined by $Q_1 = Q_2 + Q_R$ and $Q_1 S_1 = Q_2 S_2$, where $Q_1$ is the upper layer outflow, $Q_2$ the deep layer inflow, and $Q_R$ represents the net freshwater discharge (rivers, glacial melt, and precipitation).

If one applies this balance to the authors' results, where $Q_1 = A_1 c_1 \approx 22,000$ m$^3$ s$^{-1}$, the implied freshwater input ($Q_R = Q_1(S_2 - S_1)/S_2$) would range between 2,000 and 5,000 m$^3$ s$^{-1}$ (assuming typical $S_1, S_2$ gradients for the area), which appears implausibly large relative to the AF catchment and typical regional freshwater inputs; the authors should reconcile this with discharge estimates and revise the transport scaling using advective velocities instead of phase speeds and provide the specific references supporting their choice of parameters.

- The manuscript concludes that atmospheric forcing is essential for "ventilating the deep basin." However, the observational evidence provided (stations A1 and A2) is restricted to approximately 30 m depth near the shallow head of the fjord. Since the Almirantazgo Fjord reaches depths greater than 200 m, characterizing oxygen pulses at 30 m as evidence of basin-wide deep-water renewal is an overstatement. The authors must distinguish between the ventilation of the shallow sub-basin at the head and the renewal processes of the truly deep layers of the main basin, as the current generalization lacks vertical observational support.

**Minor Comments**

There are several instances where the values reported in the text are inconsistent with the underlying physics and likely represent clerical errors:

- **Wedderburn $\Delta\rho$:** On line 653, the authors cite $\Delta\rho = 3 \times 10^{-3}$ kg $m^{-3}$ for the $W_b$ calculation. This value is physically unrealistic for a stratified fjord and contradicts the mooring data showing $\Delta\rho > 10$ kg $m^{-3}$. Back-calculation suggests the intended value was likely $\sim 3.1$ kg $m^{-3}$, and the text should be corrected accordingly.

- **Seasonal Trends:** The reported negative trend for Conservative Temperature of $-1.212$°C $d^{-1}$ (line 425) is impossible given the 5°C range over five months. It appears the authors may have reported a monthly rate as a daily one. Salinity and oxygen trends show similar discrepancies.

- The use of instrumented Southern elephant seals is a major highlight and a key contribution of this study. While Figure A2 in the Appendix provides a comparison with other data sources, this figure is not discussed in the main text, nor does it allow for a direct validation of the CTD-SRLD accuracy specifically within the Almirantazgo Fjord using conventional sensors. It would be highly beneficial for the scientific community if the authors included a brief description of the quality control filters applied and quantified the deviation of the seal-borne data relative to the in-situ conventional measurements. Such a discussion is necessary to assess the reliability of these sensors in resolving the vertical stratification patterns of the fjord.

- In Sect. 4.1, the reported transport is internally inconsistent: the text states $A_1 = 3 \times 10^4$ m$^2$ and $c_1 = 1.1$ m s$^{-1}$, but then reports $Q_1 = A_1 c_1 = 2.2 \times 10^4$ m$^3$ s$^{-1}$. Arithmetically, $3 \times 10^4 \times 1.1 = 3.3 \times 10^4$ m$^3$ s$^{-1}$, so the numbers (and/or the stated $A_1$) should be corrected.

- Line 450-455: Correct "ang-fjord" to "along-fjord".

- Ensure Figure 4 labels are consistent between the map and the vertical sections.

**Summary Recommendation**

The study presents a unique dataset, but a major revision is required to correct numerical inconsistencies and ensure that the scaling, mass balance, and ventilation arguments are physically

sound.